The use of statistical and machine learning tools to accurately quantify the energy performance of residential buildings

Ibrahim Dina M. 1 2 dina.mahmoud@f-eng.tanta.edu.eg
Almhafdy Abdulbasit 3
Al-Shargabi Amal A. 1
Alghieth Manal 1
Elragi Ahmed 4
Chiclana Francisco 5
1 Department of Information Technology, College of Computer, Qassim University , Buraydah, Qassim , Saudi Arabia
2 Department of Computers and Control Engineering, Faculty of Engineering, Tanta University , Tanta , Egypt
3 Department of Architecture, College of Architecture and Planning, Qassim University , Buraydah, Qassim , Saudi Arabia
4 Department of Civil Engineering, College of Engineering, Qassim University , Buraydah, Qassim , Saudi Arabia
5 Institute of Artificial Intelligence (IAI), Faculty of Technology, De Montfort University Leicester , Leicester, Leicester , United Kingdom
Ashraf Imran
Electronic publication date: 2022 Jan 26
Publication date: 2022
Volume: 8
Electronic Location ID: e856
Received 2021 Oct 14; Accepted 2021 Dec 29
Copyright: © 2022 Ibrahim et al.
Copyright year: 2022
Copyright holder: Ibrahim et al.
License: This is an open access article distributed under the terms of the Creative Commons Attribution License, which permits unrestricted use, distribution, reproduction and adaptation in any medium and for any purpose provided that it is properly attributed. For attribution, the original author(s), title, publication source (PeerJ Computer Science) and either DOI or URL of the article must be cited.
License URL: https://creativecommons.org/licenses/by/4.0/

Keywords: Buildings characteristics, Cooling load, Heating load, Energy consumption, Statistical analysis

Funding: Qassim University coc–2019-2-2-I-5422 Qassim University, represented by the Deanship of Scientific Research, provided the financial support for this research under the number (coc-2019-2-2-I-5422) during the academic year 1440 AH/2019 AD. The funders had no role in study design, data collection and analysis, decision to publish, or preparation of the manuscript.

==============================
Prediction of building energy consumption is key to achieving energy efficiency and sustainability. Nowadays, the analysis or prediction of building energy consumption using building energy simulation tools facilitates the design and operation of energy-efficient buildings. The collection and generation of building data are essential components of machine learning models; however, there is still a lack of such data covering certain weather conditions. Such as those related to arid climate areas. This paper fills this identified gap with the creation of a new dataset for energy consumption of 3,840 records of typical residential buildings of the Saudi Arabia region of Qassim, and investigates the impact of residential buildings’ eight input variables (Building Size, Floor Height, Glazing Area, Wall Area, window to wall ratio (WWR), Win Glazing U-value, Roof U-value, and External Wall U-value) on the heating load (HL) and cooling load (CL) output variables. A number of classical and non-parametric statistical tools are used to uncover the most strongly associated input variables with each one of the output variables. Then, the machine learning Multiple linear regression (MLR) and Multilayer perceptron (MLP) methods are used to estimate HL and CL, and their results compared using the Mean Absolute Error (MAE), the Root Mean Square Error (RMSE), and coefficient of determination (R2) performance measures. The use of the IES simulation software on the new dataset concludes that MLP accurately estimates both HL and CL with low MAE, RMSE, and R2, which evidences the feasibility and accuracy of applying machine learning methods to estimate building energy consumption.

Introduction

Research on building energy consumption is motivated by the recently growing concerns on energy waste and its negative impact on the environment. When designing efficient buildings, it is essential to calculate their cooling load (CL) and heating load (HL) in order to specify the required cooling and heating equipment to achieve comfortable indoor air conditions. Architects and building designers require information about building characteristics, conditioned spaces (occupancy and activity level), climate, and intended usage (residential, industrial) to estimate the CL and HL of the building. Buildings have five distinct characteristics: environment, utilities, community, occupants, and building system (Wang et al., 2017). The environmental characteristics of a building are among the main aspects or conditions that can affect its energy consumption, i.e. contribute to sustainability and energy efficiency. Therefore, this study focuses on buildings characteristics such as wall envelope, window, and orientation.

In the literature, buildings’ characteristics have been described as “variables” (Tsanas & Xifara, 2012), “forms” (Li et al., 2019), “components” (Geyer & Singaravel, 2018), “shapes and characteristics” (Ciulla et al., 2019), and “features” (Seyedzadeh et al., 2019). Physical and non-physical factors can be used to categorize the characteristics of buildings. A window to wall ratio, for example, is a physical element of a building that is related to size, while glazing properties (e.g. U-value) are an example of physical elements of a building that are related to materials. The orientation of a building, which is determined by the cardinal and intercardinal building directions, is an example of non-physical factors.

The building characteristics in related studies can be categorized into five groups: wall variables, glazing variables, roof variables, form variables, and orientation. Glazing variables are a major architectural elements that identify the building’s features and they have a significant impact on energy performance (Tien Bui et al., 2019; Yeom et al., 2020). Five different building envelope parameters have been used to address glazing: area, area distribution, window to wall ratio (WWR), window to ground ratio (WGR), and U-value. Furthermore, when looking at each variable separately, orientation is the variable most investigated in AI research studies. Most buildings’ energy prediction studies, such as Yeom et al. (2020), Moayedi et al. (2019), Navarro-Gonzalez & Villacampa (2019), Seyedzadeh et al. (2019) and Sadeghi et al. (2020), conducted their experiments on a dataset, created by Tsanas & Xifara (2012) of 768 records and eight characteristics (relative compactness, surface area, wall area, roof area, overall height, orientation, glazing area, and distribution) used as predictors to estimate the energy consumption of the buildings.

In addition, a small number of other studies have used larger datasets. To name some of them (Himeur et al., 2020a), reviewed and examined thirty-one existing datasets based on various features such as geographical locations and rate sampling. The authors proposed a novel dataset, namely, Qatar University dataset which can be useful for any future training or testing anomaly detection algorithms. Another future direction of applying the datasets in several utilizations such as machine learning was also proposed. In addition, Li et al. (2020) used 539, 42, and 153 datasets of residential buildings, residential blocks and public buildings respectively. The authors highlighted the buildings key determinants that affect the urban building energy usage, e.g. orientation, height to canyon width perimeter-to-area ratio. (Xu & Chen, 2020), collected datasets of energy consumption from various houses in British Columbia, Canada, for 2 years. The aim was to detect anomaly energy performance in buildings. (Pham et al., 2020), used five datasets from five buildings of 1 year with an hourly resolution of energy consumption for evaluating ML-based energy prediction model. By utilizing the historical datasets, Random Forests showed good accuracy in energy prediction. (Himeur et al., 2020c), validated a recognition system based on a non-intrusive appliance model using resampled data recording in power consumption with 30,000 patterns length. The proposed model showed high accuracy in appliance recognition performance.

However, all the above mentioned studies were not constructed based on the building characteristics which emerges the gab in the existing buildings envelope based datasets. (Himeur et al., 2020b) has stated that the lack of real or well-validated datasets is one of the main obstacles that stand before anomaly prediction and detection of energy consumption in buildings. Highlighting energy output has gone through various investigations, and yet, there are still difficulties in identifying the energy performance pattern, abnormalities. Thus, this study creates a new dataset of 3,840 typical family houses in the Qassim region of Saudi Arabia, and corresponding eight characteristics to predict energy consumption, which is to be available online for public use.

Based on the created dataset, a number of classical and non-parametric statistical tools are first used to uncover the most strongly characteristics (input variables) with HL and CL (output variables). Then, two machine learning methods, the Multiple linear regression (MLR) and the Multilayer perceptron (MLP), are used to estimate HL and CL, and their results are compared using the Mean Absolute Error (MAE), the Root Mean Square Error (RMSE), and coefficient of determination (R2) performance measures. The use of the IES<VE> simulation software on the new dataset concludes that MLP accurately estimates both HL and CL with low MAE, RMSE, and R2, which evidences the feasibility and accuracy of applying machine learning methods to estimate building energy consumption. Thus, the main contributions of this study are: A new dataset of 3,840 arid climate residential buildings and corresponding eight characteristics to predict energy consumption is made publicly available.

In silico experiments on the developed dataset evidence feasibility and accuracy of applying machine learning methods to estimate building energy consumption.

The remaining of the paper is structured as follows. “Existing Datasets of Energy Consumption In Residential Buildings” presents an overview of the existing dataset used in the literature. “Methodology” details the methodology implemented to create and analyze the new dataset. “Methods and Statistical Analysis Results” reports on the results of the dataset analysis using both statistical methods and machine learning methods. “Results and Discussions discusses the obtained results and finally “Concluding Remarks and Future Research Directions” concludes the paper.

Existing datasets of energy consumption in residential buildings

The application of machine learning on building energy prediction is extensively addressed in the literature (Zhang et al., 2021). However, most of these studies focus on the algorithm implemented, while the dataset used is often overlooked. In Tsanas & Xifara (2012), Tsanas and Xifara presented a dataset on eight building characteristics (input variables X1–X8): Surface Area, Overall Height, Roof Area, Relative Compactness, Wall Area, Distribution of Glazing Area, Orientation, Area of Glazing; as predictors of buildings’ energy consumption target variables (Y1–Y2): Heating Load and Cooling Load.

Many researches have used the Tsanas & Xifara (2012) dataset for various energy prediction models in various regions using 12 different building shapes simulated in Autodesk Ecotect Analysis too (see Table 1). Kumar, Pal & Singh (2018) used the data for residential buildings in California, Navarro-Gonzalez & Villacampa (2019) in Alicante, Spain, and Roy et al. (2020) in Athens, Greece. Ciulla et al. (2019) employed nonresidential building simulation data from seven countries: Germany, Spain, the United Kingdom, Belgium, Italy, France, and Sweden. D’Amico et al. (2019) conducted research for the ANN energy assessment model on five climate zones. These studies are based on simulated data and use Tsanas and Xifara’s dataset for the training of their AI-based prediction models, i.e., machine or deep learning, as well as for testing them.

Table 1 A summary of data regarding previous studies in residential building.

References	Building characteristics	Type of energy	Building type	Location	Dataset size	
D’Amico et al. (2019)	Wall area	Energy consumption	Residential	U.S. Midwest	973	
	Wall U-value					
	Glazing area					
	Glazing U-value					
Cerquitelli, Malnati & Apiletti, 2019			Residential	Athens, Greece	768	
Ciulla et al. (2019)			Residential	N/A		
Chen & Tan (2017)	Height		Residential	N/A		
Li et al. (2019)	Relative compactness		Residential	Alicante, Spain		
Le et al. (2019a)	Wall area		Residential	Irvine		
Naji et al. (2016)	Surface area	Heating load	Residential	Greece, Athens		
Ngo (2019)	Roof area		Residential	Athens, Greece		
Kumar, Pal & Singh (2018)	Glazing area distribution	Cooling load	N/A	N/A		
Nilashi et al. (2017)	Glazing area		Residential	Ho Chi Minh City, Viet Nam		
Sadeghi et al. (2020)	Orientation		Residential	NM		
Sharif & Hammad (2019)			Residential	N/A		
Geyer & Singaravel (2018)			Residential	California		
Gao et al. (2019)	Relative compactness	Heating load	N/A	N/A	837	
Tien Bui et al. (2019)	Surface Area		Prototype model	Vietnam		
Cecconi, Moretti & Tagliabue, 2019	Wall U-value	Energy Consumption	Residential	Istanbul	180	
	Wall thickness					
	(5 different walls)					
Navarro-Gonzalez & Villacampa (2019)	Relative compactness					
	Glazing area		Residential			
	Glazing area distribution	Heating load		Athens, Greece	768	
	Roof area		Office			
	Overall Height	Cooling load			++	
	Orientation					
	Glazing area		Others			
	Glazing area distribution					
Le et al. (2019b)	Insulation K-value				Two datasets:	
	Insulation thickness	Dataset 1:				
	Wall type				180	
	Relative compactness	Energy Consumption				
	Surface area				+	
	Wall area		Residential	Istanbul, Turkey		
	Roof area				768	
	Overall height	Dataset 2:				
	Orientation				=	
	Glazing area	Cooling load				
	Glazing distribution				948	

Methodology

Sample building

There is currently a rapid construction development of residential buildings in the Qassim region. Accordingly, the Ministry of housing in Saudi Arabia launched a program of 381 villas in Buraydah city and 340 in Unayzah city, all with the same design plan. Since this is a typical new detached house in many towns in the Qassim region, it was selected and used in this study. The house plan is used in the IES<VE> simulation software. The architecture layout of the ground floor and first floor are shown in Fig. 1, while Table 2 provides information of the house envelope construction features.

Figure 1 The architecture layout (A) ground floor and (B) first floor of the house sample.

Table 2 House envelope construction features.

House features	Description	
Location	Buraydah (Coordinates: 26°22′17.8″N 43°51′29.4″E)	
Orientation	Front elevation facing South	
Shape	Typical Square and Rectangular combination of spaces	
Celling Height	3 m	
Floor Area	118.1 m2 (Ground Floor); 66.4 m2 (First Floor)	
Window Wall Ratio	10–15%	
Exterior Walls	15 mm Plaster (Dense) + 10 mm Cement + 200 mm Concrete Block (Medium) + 10 mm Cement + 15 mm Plaster (Lightweight)	
Roof	10 mm Ceramic tiles + 30 mm Concrete layer + 10 mm Extruded Polystyrene + 150 mm Reinforced Concrete (Dense)	
Windows	4 mm Double clear glass	

Modeling in IES<VE>

The IES<VE> simulation software was used to model the house for data generating (IESVE, 2008). The aim of this phase is to generate the data of the building envelope variables to analyze their effect on the building energy performance. As the building is located in Qassim, Saudi Arabia, the corresponding regional weather data file (epw. format) was imported to the software and used in the simulations. The simulation of design variables was restricted to the house’s main spaces subjected to air-conditioning, highlighted in orange in Fig. 2. Other spaces of the house, such as WC, staircase and kitchen, highlighted in blue color in Fig. 2, which are not fully air-conditioned were excluded in the simulation. The specifications of the design variables are provided in Tables 3 and 4. All thermal properties for glazing, roof and walls were carefully defined in the IES<VE> simulation software based on their U-value (Table 4), which considered the most effective property that affect the building elements’ thermal behavior. Furthermore, properties of the building elements such as doors, window frame and floors were kept constant in the IES<VE> for the simulation.

Figure 2 House sample modeling in IESVE with Qassim weather station (Sun-path).

Table 3 Air-conditioned spaces information generated from IES<VE>.

Space ID	Space name (Real)	Max. height (m)	Volume (m3)	Floor area (m2)	Floor perimeter (m)	Ext. wall area (m2)	Ext. window area (m2) 10%	
SP00000C	Bed room	5.6 (2nd f)	81.026	28.938	21.94	40.945	4.095	
SP000000	Bedroom	2.8 (1st f)	65.414	23.362	19.5	27.303	2.73	
SP000002	Living Room	2.8 (1st f)	51.614	18.433	17.18	12.183	1.218	
SP000003	Dining room	2.8 (1st f)	64.67	23.097	19.38	31.531	3.153	
SP000009	Guest room	2.8 (1st f)	81.026	28.938	21.94	40.945	4.095	
SP000005	Bed room	5.6 (2nd f)	64.67	23.097	19.38	43.394	4.339	
	Total	–	408.42	145.865	–	196.301	19.63	

Table 4 Descriptions of input and output variables in the model simulation.

Features	Description	Variables	
Building Size	Spaces in the house subjected to air-conditioning (highlighted in orange color in Fig. 2)	145.86 m2	
	184.53 m2	
Floor Height	This is referred to the internal ceiling height of spaces	2.8 m	
		3.0 m	
Glazing Area	Net area of windows	23.25 m2	
		24.15 m2	
		69.75 m2	
		72.46 m2	
		116.24 m2	
		120.76 m2	
		162.74 m2	
		169.07 m2	
		209.24 m2	
		217.37 m2	
Wall Area	Net area of walls	217.37 m2	
		209.24 m2	
		169.07 m2	
		162.74 m2	
		120.76 m2	
		116.24 m2	
		72.46 m2	
		69.75 m2	
		24.15 m2	
		23.25 m2	
WWR	Window to wall ratio of all the external wall that exposed to outdoor in all sides	10%	
	30%	
		50%	
		70%	
		90%	
Win Glazing U-value	Refer to thermal properties of glazing window which calculated by (W/m2K)	0.97	
	1.63	
		2.87	
		3.23	
		4.61	
		5.60	
Roof U-value	Refer to thermal properties of the covering of the specified spaces in the model calculated by (W/m2K)	0.13	
	0.22	
		0.35	
		0.47	
External Wall U-value	Wall envelope for the specified spaces calculated by (W/m2K)	0.26	
		0.34	
		0.60	
		1.03	
		1.62	
		2.11	
		2.82	
		3.34	
Cooling Load	Refer to the sensible cooling load through the space’s envelope (wall, window and roof) calculated by KWh per year	–	
Heating Load	Refer to the sensible heating load through the space’s envelope (wall, window and roof) calculated by KWh per year	–	

Input and output variables

As shown in Table 4, eight different design parameters of a typical house in the Qassim region were considered in order to generate the energy data to predict the whole building energy consumption. Table 4 included descriptions of each design parameters group with possible number of values. All these design parameters and values were applied in the IES<VE> simulation software and the energy consumption values in terms of cooling and heating consumption (output variables), respectively, were obtained as output from the simulation experiment. Building size and floor height have two different values that were constructed in the ModelIT application in the IES<VE> simulation software. The WWR applied to each building size and floor height for the whole external wall that exposed to the outdoor in all directions is also documented in Table 4. The remaining design parameters based on the U-values were carefully inserted in APACHE application in the IES<VE> simulation software.

As mentioned earlier, all the design parameters were applied to the main spaces only (Table 3) to ensure more reliable and accurate energy data for energy prediction. A total of 3,840 data series were introduced and simulated in the IES<VE> simulation software. A snapshot of our proposed dataset is shown in Fig. 3. Table 5 illustrates the descriptive statistics of the generated data: minimum, maximum, mean, standard deviation, variance, and skewness values.

Figure 3 A snapshot of our proposed dataset.

Table 5 Statistical descriptive of the IES<VE> simulation software generated dataset.

Features	Descriptive index	
Count	Minimum	Maximum	Mean	Std. deviation	Variance	Skewness	
Statistic	Std. Error	Statistic	Std. Error	
Building Area m2	3,840	145.86	184.53	165.2	0.31	19.33	373.94	0.0	0.04	
Floor Height m	3,840	2.8	3.0	2.9	0.0016	0.10	0.01	0.0	0.04	
Glazing Area	3,840	19.63	217.37	110.07	1.018	63.11	3,983.73	0.066	0.04	
Wall Area	3,840	19.63	217.37	110.07	1.018	63.11	3,983.73	0.066	0.04	
WWR %	3,840	10	90	50.00	0.456	28.288	800.20	0.0	0.04	
Win U-value (W/m2K)	3,840	0.97	5.60	3.16	0.025	1.59	2.56	0.150	0.04	
Roof U-value (W/m2K)	3,840	0.13	0.47	0.29	0.002	0.128	0.017	0.130	0.04	
Wall U-value (W/m2K)	3,840	0.26	3.34	1.51	0.018	1.085	1.179	0.394	0.04	
Cooling (KWh/m2. yr)	3,839	5.45	671.60	336.85	2.18	135.31	18,309.8	0.239	0.04	
Heating (KWh/m2. yr)	3,839	0.0	7.03	0.95	0.02	1.31	1.701	1.892	0.04	

Methods and statistical analysis results

This section analyses first the main statistical properties of the variables of the new dataset with the help of histograms and scatterplots. Then, the relationship between the input and output variables is analyzed using the Spearman rank correlation coefficient. Finally, our dataset is analyzed using two machine learning approaches, the Multilayer Regression (MLR) and Multilayer Perceptron (MLP) methods, respectively.

Data exploration

The simulated buildings were generated using the IES<VE> simulation software for Buraydah city. The Qassim province was chosen as it has a hard-arid climate with exceptionally hot summers and cool winters, requiring a lot of energy for cooling and heating residential buildings. The dataset is available at Almhafdy (2021) and contains 3,840 records. The following nine constant characteristics were used: location (Buraydah), orientation (front façade oriented to south), shape (rectangular and square spaces), ceiling height (3 m), floor area (ground floor 118.1 m2; first floor 66.4 m2), window wall ratio (10–15%), Exterior walls (0.015 m plaster + 0.01 mm cement + 0.020 m concrete block (medium) + 0.01 m cement + 0.015 m plaster (lightweight), roof (0.01 m ceramic tiles + 0.03 m concrete layer + 0.01 m extruded polystyrene + 0.015 m reinforced concrete, and windows (0.004 m double clear glass).

Two building sizes were used 145.86 m2 and 184.53 m2. For each building size two floor heights of 2.8 m and 3 m were used; five different WWR as percentage of all external wall exposed to outdoor were used: 10%, 30%, 50%, 70%, and 90%; six win-value were simulated: 0.97, 1.63, 2.87, 3.23, 4.61, and 5.60); four different roof U-value were simulated: 0.13, 0.22. 0.35, and 0.47; and eight wall U-value were applied to each roof U-value. This is illustrated in Fig. 4.

Figure 4 Input design parameters groups for energy consumption of building.

Accordingly, we obtained 2 * 2 * 5 * 6 * 4 * 8 = 3,840 building samples. The simulate buildings are characterized by eight building features (input variables), and their output HL and CL were recorded, as summarized in Table 6.

Table 6 Mathematical representation of the input and output variables with the number of possible values.

Mathematical representation	Input or output variable/Feature	No. of possible values	Label for charts	
I1	Building Size	2	BA	
I2	Floor Height	2	FH	
I3	Glazing Area	10	GA	
I4	Wall Area	10	WA	
I5	WWR	5	WWR	
I6	Win Glazing U-value	6	WinU	
I7	Roof U-value	4	RU	
I8	External Wall U-value	8	WU	
O1	Cooling Load	3,659	CL	
O2	Heating Load	2,674	HL	

Statistical properties of the variables were first analyzed with visualization of the empirical probability distributions of all the input and output variables (Tsanas & Xifara, 2012). These are provided in Fig. 5 which presents the probability density estimates using histograms of the output variable: the cooling load and the heating load. Figure 5A shows the frequency distribution for the cooling load output variable that resulted in the 3,840 records in the dataset and it describes that the most values are within a range of 100 to 600. While in Fig. 5B, the frequency distributions show that most of the values of the output variable heating load are ranged between 0.0 to 0.2. As a result, the necessity to experiment with machine learning approaches such as multiple linear regression (MLR) and multilayer perceptron (MLP) is intuitively justified.

Figure 5 Probability density estimates using histograms of the output variable (A) cooling load, and (B) heating load.

Statistical analysis

Due to the general non-Gaussian nature of the data, the Spearman rank correlation coefficient was used to derive a statistical metric for the strong relationship between each input variable with each of the two output variables (Tsanas & Xifara, 2012), which is given in Table 7. It is evident that several of the input variables are highly associated, such as GA (Glazing Area) and WWR (Window to Wall Ratio). As it is naturally expected, the variables GA and WWR are almost inversely proportional to WA.

Table 7 Correlations matrix using Spearman rank correlation between the eight input variables.

	BA	FH	GA	WA	WWR	WinU	RU	WU	
BA	1.000	0.000	0.173	0.173	0.000	0.000	0.000	0.000	
FH	0.000	1.000	0.087	0.087	0.000	0.000	0.000	0.000	
GA	0.173	0.087	1.000	−0.925	0.981	0.000	0.000	0.000	
WA	0.173	0.087	−0.925	1.000	−0.981	0.000	0.000	0.000	
WWR	0.000	0.000	0.981	−0.981	1.000	0.000	0.000	0.000	
WinU	0.000	0.000	0.000	0.000	0.000	1.000	0.000	0.000	
RU	0.000	0.000	0.000	0.000	0.000	0.000	1.000	0.000	
WU	0.000	0.000	0.000	0.000	0.000	0.000	0.000	1.000	
Note:

BA: Building Size, FH: Floor Height, GA: Glazing Area, WA: Wall Area, WWA: Window to Wall Ratio, WinU: Win Glazing U-value, RU: Roof U-value, WU: External Wall U-value.

We can similarly depict the bivariate correlations between the eight input variables using a scatter plot matrix. A scatter plot matrix is a grid (or matrix) that represents a single view with multiple scatterplots in a matrix format (Elmqvist, Dragicevic & Fekete, 2008). Each scatter plot in the matrix depicts the relationship between two variables, allowing for the exploration of multiple relationships in a single graph. Figure 6 shows a scatter plot matrix of our eight input variables. The position of each dot on the horizontal and vertical axis indicates values for an individual data point. For each pairwise combination of variables chosen, a scatter plot is constructed.

Figure 6 Scatter plot matrix representation of the eight input variables.

Machine learning-based analysis

The main objective of this study is to describe a dataset generated for the energy consumption of buildings in the arid climate. This section makes use of two machine learning models, namely Multiple Linear Regression (MLR) and Multilayer Perceptron (MLP). These two models were chosen to examine the viability of the developed dataset in predicting the buildings energy consumption in terms of cooling and heating loads. In a recent study of ours (Al-Shargabi et al., 2021), we applied deep learning and created various models to predict the energy consumption of buildings using the dataset described in this study.

Multiple linear regression analysis

Multiple regression extends simple linear regression to predict the value of a variable (the outcome, target or criterion variable) based on the values of two or more other variables (the predictor, explanatory or regressor variables) (Tian et al., 2017).

This section examines the distribution of the output variables (CL and HL) using the normal P–P plot, and the scatter plot of the regression standardized residual. The normal P–P plot of the standardized residual for dependent variables CL and HL is shown in Fig. 7, which corroborates that CL is normally distributed while HL is not.

Figure 7 The normal P–P plot of the regression standardized residual for our dependent variables CL and HL.

Cross validation (CV) is a common statistical re-sampling technique used in this paper. The dataset is divided into two subsets: a training subset and a testing subset. The training subset is used to derive model parameters, while the testing subset is used to compute errors (out-of-sample error or testing error). In particular, 10-fold CV (Uyank & Güler, 2013) is used as the learner testing method. We investigate how accurate the actual statistical mapping is reporting out-of-sample errors after conducting the exploratory statistical analysis, which provides important insight into the strength of the association between the input parameters and the output variables. The mean value of each MLR coefficient over the 10-fold CV iterations is obtained and used for predicting CL and HL in Eqs. (1) and (2), respectively.

(1) MLRCL=11.448−3.24I1−75.083I2+2.468I3+3.313I4+5.519I5+34.84I6+37.093I7+29.89I8

(2) MLRHL=−0.029−0.035I1−1.209I2+0.035I3+0.035I4+0.004I5+0.401I6+0.795I7+0.508I8

Multilayer perceptronanalysis

In this model, using our proposed dataset, an ANN using the Multilayer perceptron method, which is one of the most commonly used methods for building an ANN (Hastier, Tibshirani & Friedman, 2009), is built in SPSS.

Artificial neural networks (ANN) are nonlinear models that fall into the artificial intelligence technique category known as black-box models (Heddam, 2016). The multilayer perceptron neural network (MLP) (Rumelhart, Hinton & Williams, 1985) is one of the most extensively used ANN architectures in the literature, and it is extensively employed in hydrological, water resources, and environmental applications. Three layers make up the MLP: the input layer contains the independent variables, the output layer contains the dependent variable, and one or more hidden layers may also be present. The parameters of the MLP model are its weights and biases. It was used to alter the weights and biases of the training subsets, and the MLP was then trained with random beginning values. To choose the model with the lowest MSE between actual and predicted CL and HL, the training process is repeated many times. Neural networks with Sigmoid activation functions in their hidden layers and linear activation functions in their output layers, commonly known as the identity function, are employed for this research.

To select the number of hidden layers, automatically architecture selection is chosen. The following three different distributions for the dataset are applied: (i) 70% to train the NN and 30% to test the NN; (ii) 80% to train the NN and 20% to test the NN; (iii) 90% to train the NN and 10% to test the NN. Figures 8 and 9 show the obtained NNs to predict CL and HL from the set of 8 input variables, respectively.

Figure 8 Multilayer perceptron model for predicting the cooling load output from the input variables.

Figure 9 Multilayer perceptron model for predicting the heating load output from the input variables.

The importance score of each of the eight independent variables in the prediction of each of the output variables is computed and given as Table 8. According to Table 8, the top five important input variables when predicting both the CL and HL output variables are WWR, WinU, GA, WA, and WU. Figures 10 and 11 shows the importance distribution percentages of the input variables as determined by the MLP for the CL and HL output variables, respectively. The top five important input variables are further investigated in terms of their effect on predicting buildings energy consumption in “Concluding Remarks and Future Research Directions”. These five input variables are the base to create various combinations to several prediction models of the CL and HL.

Figure 10 Importance distribution of the input variables as determined by the MLP for the cooling load output variables.

Figure 11 Importance distribution of the input variables as determined by the MLP for the heating load output variables.

Table 8 Importance of the input variables as determined by the MLP for the output variables.

Measure	Importance score with CL	Importance score with HL	
BA	0.049 ± 0.015	0.067 ± 0.025	
FH	0.024 ± 0.003	0.023 ± 0.009	
GA	0.209 ± 0.129	0.087 ± 0.033	
WA	0.126 ± 0.028	0.111 ± 0.040	
WWR	0.240 ± 0.142	0.157 ± 0.031	
WinU	0.230 ± 0.041	0.296 ± 0.009	
RU	0.015 ± 0.002	0.038 ± 0.002	
WU	0.108 ± 0.018	0.252 ± 0.009	

Error and performance measures

This section reports on the general performance of the trained methods that were discussed in the previous section. The models are compared using three performance measures, namely, Mean Absolute Error (MAE), Root Mean Square Error (RMSE) and coefficient of determination (R2).

The average difference between expected and actual variables, such as heating and cooling loads, is known as the Mean Absolute Error (MAE). In (Eq. 3), the following equation demonstrates how MAE is calculated:

(3) MAE=(1/n)×∑i=1n|pi−yi|

Prediction errors are calculated by calculating the Root Mean Square Error (RMSE). Large variations between expected and actual results can be captured using this method. The lower the RMSE, the more accurate the model is. In (Eq. 4), the RMSE is determined using the following equation:

(4) RMSE=(1/n)×∑i=1n[pi−yi]2

The coefficient of determination (R2) indicates how much of the variance in the dependent variable can be predicted using the independent variables, such as heating and cooling loads. The closer value to 1, the higher performance model and the stronger relationship, as calculated in (Eq. 5).

(5) R2=∑i=1n(pi−y¯)2∑i=1n(yi−y¯)2

where pi identifies the predicted value for sample i, yi identifies the actual value for sample i, n is the sample size, y¯ indicates the mean of the predicted values.

Results and Discussions

This study investigated various combinations of the eight building characteristics variables as inputs to the MLP and MLR models in order to examine the effect of these variables on the energy consumption in terms of heating and cooling loads. During this research, a total of eight different models were created and compared (Tables 9 and 10).

Table 9 Out of sample MAE, RMSE, and R2 for predicting the CL output variable for the MLR and MLP models.

Model		Input variables	MAE	RMSE	R2	
MLP	M1	WinU+WWR+WU+GA+WA	23.2	42.92	0.999	
	M2	WinU+WWR+WU+GA+WA+BA	23.05	41.54	0.999	
	M3	WinU+WWR+WU+GA+WA+FH	22.71	38.69	0.997	
	M4	WinU+WWR+WU+GA+WA+RU	22.88	40.14	0.998	
	M5	WinU+WWR+WU+GA+WA+BA+FH	22.39	35.7	0.995	
	M6	WinU+WWR+WU+GA+WA+BA+RU	22.51	37.22	0.996	
	M7	WinU+WWR+WU+GA+WA+FH+RU	22.07	32.56	0.993	
	M8	All: WinU+WWR+WU+GA+WA+BA+FH+RU	21.78	29.123	0.992	
MLR	M1	WinU+WWR+WU+GA+WA	47.91	66.32	0.990	
	M2	WinU+WWR+WU+GA+WA+BA	46.97	61.40	0.984	
	M3	WinU+WWR+WU+GA+WA+FH	47.37	63.56	0.986	
	M4	WinU+WWR+WU+GA+WA+RU	47.62	64.87	0.988	
	M5	WinU+WWR+WU+GA+WA+BA+FH	46.26	57.43	0.979	
	M6	WinU+WWR+WU+GA+WA+BA+RU	46.66	59.70	0.982	
	M7	WinU+WWR+WU+GA+WA+FH+RU	46.38	58.15	0.980	
	M8	All: WinU+WWR+WU+GA+WA+BA+FH+RU	46.020	56.015	0.978	

Table 10 Out of sample MAE, RMSE, and R2 for predicting the HL output variable for the MLR and MLP models.

Model		Input variables	MAE	RMSE	R2	
MLP	M1	WinU+WWR+WU+GA+WA	0.180	0.376	1	
	M2	WinU+WWR+WU+GA+WA+BA	0.177	0.346	1	
	M3	WinU+WWR+WU+GA+WA+FH	0.179	0.368	1	
	M4	WinU+WWR+WU+GA+WA+RU	0.175	0.333	0.996	
	M5	WinU+WWR+WU+GA+WA+BA+FH	0.174	0.320	0.992	
	M6	WinU+WWR+WU+GA+WA+BA+RU	0.170	0.284	0.981	
	M7	WinU+WWR+WU+GA+WA+FH+RU	0.172	0.308	0.989	
	M8	All: WinU+WWR+WU+GA+WA+BA+FH+RU	0.167	0.260	0.433	
MLR	M1	WinU+WWR+WU+GA+WA	0.955	1.567	0.469	
	M2	WinU+WWR+WU+GA+WA+BA	0.942	1.455	0.521	
	M3	WinU+WWR+WU+GA+WA+FH	0.948	1.510	0.493	
	M4	WinU+WWR+WU+GA+WA+RU	0.945	1.481	0.507	
	M5	WinU+WWR+WU+GA+WA+BA+FH	0.935	1.399	0.552	
	M6	WinU+WWR+WU+GA+WA+BA+RU	0.921	1.269	0.627	
	M7	WinU+WWR+WU+GA+WA+FH+RU	0.928	1.337	0.587	
	M8	All: WinU+WWR+WU+GA+WA+BA+FH+RU	0.915	1.223	0.656	

According to testing data, the MAE, RMSE, and R2 statistics of several MLP and MLR models in predicting the cooling load (CL) are shown in Table 9. Table 9 shows significant differences across the eight MLP models based on the three performance indicators. Between 21.78 and 23.2 (MAE, RMSE, and R2), respectively, the values of MAE, RMSE, and R2 were found. MAE and RMSE performance metrics have the lowest values when all eight of the building’s identifying attributes are supplied into the M8 model (WinU, WWR, WU, GA, WA, BA, FH, and RU). The highest R2 values were found with the M1 and M2 models, however, the M8 model still had the highest value. As can be seen from the results, the MLP M8 model has excellent cooling load (CL) performance and outstanding overall accuracy in predicting cooling load.

Table 9 also displays the results of the cooling load (CL) prediction using MLR models based on the testing data. The MAE and RMSE metrics based on MLR models yield poorer results than those based on MLP models. Furthermore, the eight MLR models revealed considerable variances depending on the three performance measurements criterion, as shown in Table 9. MAE, RMSE, and R2 values varied from 46.02 to 47.91, 56.01 to 66.32, and 0.978 to 0.99, respectively. The M8 model, which employs all eight building characteristics variables as input, also yields the lowest values of the MAE and RMSE performance measures (WinU, WWR, WU, GA, WA, BA, FH, and RU). The highest values for the R2 measure were obtained with the M1 model, which was not far off from the value obtained with the M8 model. In terms of MAE, RMSE, and R2 statistics, Table 9 compares the effectiveness of several MLP and MLR models in forecasting cooling load (CL).

Similarly, Table 10 reported the results obtained in predicting the heating load (HL) based on the same three performance measures. The MAE, RMSE, and R2 values for the MLP models ranged from (0.167 to 0.18), (0.26 to 0.37), and (0.43 to 1.00), respectively, according to Table 10. The M8 model, which employs all eight building characteristics variables as input, likewise produces the lowest MAE and RMSE performance scores (WinU, WWR, WU, GA, WA, BA, FH, and RU). With the M1, M2, and M3 models, the highest R2 values were found. The MAE and RMSE figures indicate that the MLP model’s performance is extremely good, and the MLP M8 model generally achieves good forecast accuracy of heating load (HL). Table 10 also displays the heating load (HL) prediction results derived using MLR models based on the testing data. The MAE and RMSE values based on the MLR models are lower than those based on the MLP models, as evidenced by the cooling load projections in Table 9. Table 10 shows that the eight MLR models differed significantly based on the three performance measurements criterion. MAE, RMSE, and R2 values varied between (0.915 to 0.955), (1.223 to 1.567), and (0.469 to 0.656), respectively. The M8 model, which employs all eight building characteristics variables as input, also yields the lowest MAE and RMSE values and the highest value of R2 performance metrics (WinU, WWR, WU, GA, WA, BA, FH, and RU).

In comparison, the prediction accuracy of heating load (HL) for the regression models was higher than the prediction accuracy of cooling load (HL) in both MLP and MLR models for all eight generated combinations, according to the data provided in Table 9.

The comparison of the models was based on graphical plots as scatter plots, box plots, violin plots, and Taylor diagram plots. Figures 12 and 13 show the scatterplots of the actual and the predicted values of the cooling load and the heating loads output variables obtained by MLP and MLR when using all the inputs, as represented in model M8 in Table 9 and 10. The best cooling load results of R2 with 0.976 was achieved by MLP, whereas the MLR model provides R2 with 0.839. similarly, the R2 value for the heating load using MLP model is 0.958 which is better than the 0.438 R2 value given by the MLR model.

Figure 12 Scatterplots showing the relation between the actual and the predicted values of the cooling load (CL) and heating loads (HL) variables for the MLP M8 model.

Figure 13 Scatterplots showing the relation between the actual and the predicted values of the cooling load (CL) and heating loads (HL) variables for the MLR M8 model.

The violin plots and the box plots for the actual and the predicted values of the heating load and cooling load output variables are illustrated in Figs. 14–17. As in the violin plots presented in Fig. 14, the two lines with a black square and red circle color display the mean and the median values of the heating and cooling loads, respectively. The high resemblance between the actual and the predicted heating load was achieved by MLP, especially on the median (323.18 and 323.85), while the MLR median value is 339.36. While the values of the mean for the CL in the actual, predicted MLP and MLR are very close (336.85, 337.08, and 338.75), as illustrated in Table 11. Similarly, for the heating load in Fig. 15 and Table 11, the high similarity between the actual and the predicted heating load was also accomplished by MLP with median values 0.38 and 0.43 where the median of the MLR is 0.96.

Figure 14 Violin plots of the actual and the predicted values of the cooling load (CL) values obtained by MLP and MLR.

Figure 15 Violin plots of the actual and the predicted values of the heating load (HL) values obtained by MLP and MLR.

Figure 16 Box plots of the actual and the predicted values of cooling load (CL) values obtained by MLP and MLR.

Figure 17 Box plots of the actual and the predicted values of heating load (HL) values obtained by MLP and MLR.

Table 11 The mean and median values obtained by the actual, the MLP, and the MLR predicted models for the CL and HL variables derived from the violin and box plots.

Output variable	Model	Mean	Median	
		Violin plot	Box plot	Violin plot	Box plot	
CL	Calculated (simulated)	336.85	336.84	323.18	323.17	
	MLP	337.08	337.08	323.85	323.84	
	MLR	338.75	338.75	339.36	339.35	
HL	Calculated (simulated)	0.95	0.953	0.38	0.383	
	MLP	0.96	0.961	0.43	0.433	
	MLR	0.96	0.962	0.96	0.96	

Figure 16 illustrates the box plots of the actual and the predicted cooling load by MLP and MLR models. The median is represented by the central line with values 323.17, 323.84, and 339.35 for the actual, the predicted MLP, and the predicted MLR, respectively. This indicates that the MLP model is better than the MLR model, as shown in Table 11. The 25th and 75th percentiles are represented by the box’s two edges, and the x symbol represents the mean points which have values 336.85, 337.08, and 338.75 for the actual, the predicted MLP, and the predicted MLR, respectively. Likewise, Fig. 17 demonstrates the box plots of the actual and the predicted heating load variables obtained by MLP and MLR models. The median is represented by the central line with values 0.383, 0.433, and 0.96 for the actual, the predicted MLP, and the predicted MLR, respectively. It is clear from the box plots that the MLR model gives better values near the actual cooling and heating loads.

Finally, the Taylor diagram plot was used to compare the MLP and the MLR models for the cooling load and the heating load as in Figs. 18 and 19, respectively. Taylor diagram plot is one of the most and highly recommended diagrams for performance comparisons of machine learning (Zhu et al., 2019). It exhibits three specific statistics: Pearson correlation (R), ratio value, and the normalized standard deviation. The ratio value means the ratio of the normalized variances indicates the relative amplitude of the model and observed variations. It is shown from the two figures that MLP performed better than MLR. In general, the MLP points, represented by the blue circle, are closer to the reference points than the blue star symbols that signify the MLR. The ratio values for the CL variable predicted by the MLP model is 0.989 and the MLR model is 0.899. Whereas in the HL variables, the predicted MLP and the predicted MLR model gives ratio values with 0.971 and 0.622, respectively, as illustrated in Table 12. The table also represents the correlation values of the two MLP and MLR models for the CL and HL variables where the CL the MLP gives 0.988 correlation value while the MLR gives 0.911. For the HL, the MLP and the MLR correlations values are 0.979 and 0.659, respectively. These plots demonstrate that the MLP model predicts the cooling load and the heating load output variables in a better way compared to the MLR model when comparing the actual values with the predicted values.

Figure 18 Taylor diagram of the actual and the predicted cooling load (CL) values obtained by MLP and MLR.

Figure 19 Taylor diagram of the actual and the predicted heating load (HL) values obtained by MLP and MLR.

Table 12 The ratio and correlation values obtained by the MLP and MLR models for the CL and HL variables using the Taylor diagram.

Output variable	Model	Ratio value	Correlation value	
CL	MLP	0.989	0.988	
	MLR	0.899	0.911	
HL	MLP	0.971	0.979	
	MLR	0.622	0.659	

Concluding Remarks and Future Research Directions

Predicting building energy consumption is critical for achieving energy efficiency and sustainability. Nowadays, building energy simulation software is frequently used to assess or predict building energy usage to aid in the design and operation of energy-efficient buildings. This paper investigated the impact of eight input variables on residential buildings heating load (HL) and cooling load (CL), respectively. A variety of classical and non-parametric statistical analytic tools were used to find the most strongly associated input variables with each of the output variables. Then, using the performance measures Mean Absolute Error (MAE), Root Mean Square Error (RMSE), and coefficient of determination (R2), two machine learning statistical methods to estimate HL and CL were compared: Multiple linear regression (MLR) and Multilayer perceptron (MLP). Simulation experiments on 3,840 different residential buildings showed that HL and CL can accurately be predicted using the IES<VE> simulation software actual data with low MAE, RMSA, and R2 values, especially when using the MLP approach.

The findings of this study suggest that predicting building parameters using machine learning methods is a practical and accurate method. Among the major findings of this study is that the MLP models are more accurate in predicting both cooling and heating loads of the buildings, as compared to the MLR models. Also, the best performed MLP model was the one that uses the eight input variables.

Based on the eight buildings characteristics input variables, many various combinations can be created for predicting the energy consumption, however, and due to the time limitation, only eight combinations have been considered with a focus on the most important input variables.

The obtained results in this paper suggest that future research on the application of additional machine learning and deep learning models to analyze our proposed dataset and comparison with other benchmark datasets is worth considering.

Supplemental Information

Supplemental Information 1 A new dataset for residential buildings energy consumption of 3,840 records which is larger than the existing ones.

a new dataset for energy consumption of residential buildings and investigates the impact of residential buildings’ eight input variables (Building Size, Floor Height, Glazing Area, Wall Area, window to wall ratio (WWR), Win Glazing U-value, Roof U-value, and External Wall U-value) on the heating load (HL) and cooling load (CL) output variables.

Click here for additional data file.

Additional Information and Declarations

Competing Interests

Author Contributions

Data Availability

The authors declare that they have no competing interests.

Dina M. Ibrahim conceived and designed the experiments, performed the experiments, analyzed the data, prepared figures and/or tables, authored or reviewed drafts of the paper, and approved the final draft.

Abdulbasit Almhafdy conceived and designed the experiments, performed the experiments, performed the computation work, prepared figures and/or tables, authored or reviewed drafts of the paper, and approved the final draft.

Amal A. Al-Shargabi conceived and designed the experiments, performed the computation work, prepared figures and/or tables, authored or reviewed drafts of the paper, and approved the final draft.

Manal Alghieth analyzed the data, prepared figures and/or tables, authored or reviewed drafts of the paper, and approved the final draft.

Ahmed Elragi performed the experiments, prepared figures and/or tables, authored or reviewed drafts of the paper, and approved the final draft.

Francisco Chiclana analyzed the data, prepared figures and/or tables, authored or reviewed drafts of the paper, and approved the final draft.

The following information was supplied regarding data availability:

The data is available at GitHub: https://github.com/Dr-Dina-M-Ibrahim/A-dataset-for-residential-buildings-energy-consumption-with-statistical-and-machine-learning-analysi.git.

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
