# Peer review of "The use of statistical and machine learning tools to accurately quantify the energy performance of residential buildings"

_PeerJ Computer Science, doi:10.7717/peerj-cs.856_

## Round 0.1 · original submission · Major Revisions

Based on the comments from the reviewers, the authors are advised to make "major revisions" for the submitted manuscript.

·

Basic reporting

In the present work, the authors applied and compared two machines learning models for better prediction of heating load (HL) and cooling load using eight input variables namely, Building Size, Floor Height, Glazing Area, Wall Area, Window to Wall Ratio, Win Glazing U-value, Roof U-value, and the External Wall U-value. The proposed models were; the multilayer perceptron neural network (MLPNN) model and the multiple linear regression model (MLR). The two models were developed using more than 3840 building samples, and three modelling strategies were analyzed based on different splitting ratios: 70/30, 80/20, 90/10. Three performances metrics were used for models evaluation and comparison: the RMSE, MAE and R2. The investigation showed that the MLPNN was more accurate than the MLR, which is a logical finding.

The focus of the paper is very interesting, worthy of investigation and relevant to the PeerJ Journal. The necessary information’s for such study were provided by the authors and clearly presented. The writing style is good and the reader can easily understand what the authors tried to explain and to demonstrate. I have no problem about the paper structure, except some figures that I estimate they are non-value added as there are any comments of discussion about containing information in these figures (5-11). I would like to follow the positive aspect of the paper by mentioning some weak points. Indeed, the section results and discussion seems still to be the weak part of the investigation and it has significantly contributed to degrading the overall paper quality, information losses, worsened understanding of the overall results of the investigation and has not contributed to an increased paper reading. In conclusion, the paper needs in depth amendments before to be accepted.
1. Theoretical description of the MLPNN should be presented and clearly described.
2. In such modelling study, a comparison between the models having different input combinations should be conducted (at least six combinations).
3. The contribution of the input variables calculated as a percent ratio can help in better understanding the manuscript, see for example the connection weights approach (CW-AP) (Olden and Jackson 2002; Olden et al. 2004) and the Garson approach (G-AP) (Garson 1991).
4. Obtained results should be deeply discussed.
5. The scatterplot between measured and predicted data should be provided (only for the testing data).
6. The following figures are necessary: boxplot, violin plot, and Taylor diagram (only for the testing data).

Experimental design

see comments above

Validity of the findings

see comments above

Additional comments

see comments above

Reviewer 2 ·

Basic reporting

This paper presents a new building energy consumption dataset and investigates the impact of eight input variables on residential buildings heating load (HL) and cooling load (CL), respectively. A variety of classical and non-parametric statistical analytic tools were used to find the most strongly associated input variables with each of the output variables. Then, using the performance measures Mean Absolute Error (MAE), Root Mean Square Error (RMSE), and coefficient of determination (R2), two machine learning statistical methods to estimate HL and CL were compared: Multiple linear regression (MLR) and Multilayer perceptron (MLP).

The contribution of the paper is interesting and relevant to building energy research community. The proposed solution and evaluation methodology are original. However, the manuscript still can be improved through the consideration of the following aspects:

1) The authors failed to motivate their proposed work in the Introduction. The paragraph should be added to the introduction for highlighting the applications of the building energy consumption datasets, such as energy prediction, energy disaggregation (or non-intrusive load monitoring), anomaly detection, fault diagnosis, etc. To that end, I suggest to the following references when you address this comments:
- Identifying key determinants for building energy analysis from urban building datasets
- Artificial intelligence based anomaly detection of energy consumption in buildings: A review, current trends and new perspectives
- A hybrid data mining approach for anomaly detection and evaluation in residential buildings energy data
- Predicting energy consumption in multiple buildings using machine learning for improving energy efficiency and sustainability
- Robust event-based non-intrusive appliance recognition using multi-scale wavelet packet tree and ensemble bagging tree
2) The literature review is very terse and some new articles discussing building energy consumption datasets are missing, such as :
- Building power consumption datasets: Survey, taxonomy and future directions
- A novel approach for detecting anomalous energy consumption based on micro-moments and deep neural networks
Where a new dataset, called Qatar University dataset (QUD) has been presented.
3) The drawbacks and limitations of the proposed work should be discussed in the conclusion along the most important findings.
4) The paper should be carefully proofread as there are some typos and grammatical issues.

Experimental design

The experimental design is well discussed in the paper. the latter explains well the procedure used to gather data and validate the dataset with two machine learning models.

Validity of the findings

More discussions are required to highlight the drawbacks and limitations of the proposed approach, and the future directions to overcome these issues.

Reviewer 3 ·

Basic reporting

The article is clear, but requires a slight correction of the English language. Literature references are sufficient. Results supported by examples.

Experimental design

The paper is an interesting approach in terms of forecasting energy consumption. Nevertheless, in the methods based on machine learning, an essential element is the training and validation set. The effectiveness of forecasting and the thesis mainly formulated depends on selecting criteria, the amount of data, and the appropriate data selection. Therefore, the universality of the algorithm has some limitations.

Validity of the findings

1) I propose to describe in more detail on what basis and the scope of selecting input data for the given research problem results.
2) The authors could refer to other machine learning methods and justify the choice of the methods presented.

---

## Round 0.2 · accepted · Accept

Based on the reviewer's response and comments, your paper has been accepted. Authors are advised to check for typos and grammatical errors in the manuscript.

·

Basic reporting

The have significantly improved the paper and the necessaries revisions were correctly done. The paper is ready for publication, no further revision is necessary. well

Experimental design

well conducted

Validity of the findings

well presented and discussed

Additional comments

The have significantly improved the paper and the necessaries revisions were correctly done. The paper is ready for publication, no further revision is necessary.

Reviewer 2 ·

Basic reporting

The paper has been significantly improved after revision.

Experimental design

The experimental design part has been improved after revision.

Validity of the findings

The main findings of the paper are innovative.

Additional comments

The authors have addressed all my comments, I have no further suggestions.